# The Association between Childhood Obesity and Cardiovascular Changes in 10 Years Using Special Data Science Analysis

**DOI:** 10.3390/children10101655

**Published:** 2023-10-05

**Authors:** João Rala Cordeiro, Sara Mosca, Ana Correia-Costa, Cátia Ferreira, Joana Pimenta, Liane Correia-Costa, Henrique Barros, Octavian Postolache

**Affiliations:** 1Instituto de Telecomunicações, IT-IUL, Iscte—Instituto Universitário de Lisboa, 1649-026 Lisbon, Portugal; cordeirojoao@gmail.com; 2Pediatric Nephrology Unit, Centro Materno-Infantil do Norte, Centro Hospitalar Universitário de Santo António, 4099-001 Porto, Portugal; sara.mosca.silva@gmail.com (S.M.); lianecosta@icbas.up.pt (L.C.-C.); 3Division of Paediatric Cardiology, Centro Hospitalar Universitário São João, 4200-319 Porto, Portugal; analuisacrcosta@gmail.com (A.C.-C.); joanapimenta@hotmail.com (J.P.); 4EPIUnit—Instituto de Saúde Pública, Universidade do Porto, Rua das Taipas, n° 135, 4050-600 Porto, Portugal; catia.ferreira@ispup.up.pt (C.F.); henrique.barros@ispup.up.pt (H.B.); 5Laboratório para a Investigação Integrativa e Translacional em Saúde Populacional (ITR), Universidade do Porto, Rua das Taipas, n° 135, 4050-600 Porto, Portugal; 6Departamento de Ciências da Saúde Pública e Forenses e Educação Médica, Faculdade de Medicina, Universidade do Porto, 4200-319 Porto, Portugal

**Keywords:** cardiovascular risk, childhood obesity, ECG analysis, neural architecture search, 1D convolutional neural network, 1D CNN

## Abstract

The increasing prevalence of overweight and obesity is a worldwide problem, with several well-known consequences that might start to develop early in life during childhood. The present research based on data from children that have been followed since birth in a previously established cohort study (Generation XXI, Porto, Portugal), taking advantage of State-of-the-Art (SoA) data science techniques and methods, including Neural Architecture Search (NAS), explainable Artificial Intelligence (XAI), and Deep Learning (DL), aimed to explore the hidden value of data, namely on electrocardiogram (ECG) records performed during follow-up visits. The combination of these techniques allowed us to clarify subtle cardiovascular changes already present at 10 years of age, which are evident from ECG analysis and probably induced by the presence of obesity. The proposed novel combination of new methodologies and techniques is discussed, as well as their applicability in other health domains.

## 1. Introduction

Paediatric obesity in both children and adolescents constitutes a major global public health problem. Over recent decades and according to the 2020 World Health Organization (WHO) report, the prevalence of paediatric overweight and obesity rose dramatically from 4% in 1975 to 18% (over 360 million children and adolescents) in 2016. The worldwide level of obesity has nearly tripled in 2020 when compared to that in 1975 [1,2,3]. 

From birth to adulthood, individuals are susceptible to multiple influences, which can determine a higher likelihood of them becoming overweight or obese. There is evidence supporting that the metabolic and cardiovascular (CV) consequences of obesity, such as hypertension, dyslipidaemia and endothelial function impairment, with the development and progression of atherosclerosis, begin early in life and accompany the obese child throughout their life [4,5,6,7,8].

The clustering of CV risk factors in early childhood is concerning, given that ~80% of obese children remain obese in adulthood. Recently, several environmental changes have encouraged the rise in the prevalence of childhood obesity and created the vicious cycle of childhood obesity. The root causes of obesity development in childhood and adolescence reflect complex interactions among environmental, socioeconomic, behavioural and genetic factors [1,9].

There is solid evidence supporting obesity as an independent risk factor for cardiovascular disease (CVD) in adults, but the evidence is scarce in paediatrics. Nonetheless, obesity and its risk factors do not influence all individuals equally. Thus, prevention, screening and follow-ups have become a challenge for all physicians and researchers [10]. 

The cardiovascular function assessment of children and adolescents may include a standard 12-lead electrocardiogram (ECG), pulse wave velocity analysis, office blood pressure measurement and 24 h ambulatory blood pressure monitoring [11].

Studies performed on adult populations present evidence suggesting a close association between obesity and a wide spectrum of ECG abnormalities, including leftward shifts in electrocardiographic axes, left ventricular hypertrophy and the flattening of the T wave. Moreover, some of these ECG changes may be reversed with weight loss. However, the correlation between obesity and ECG in children and adolescents has not been studied in detail on a large scale, and there are hardly any concerning the correlations between body fat distribution and ECG [12,13].

In general, epidemiological research on obesity impact is usually (if not exclusively) based on standard statistical analysis where the output generally consists of univariable or/and multivariable analysis, multiple linear regression analysis, ratios, ranges, correlations, among other statistical measures. On one hand, these studies mainly address the risk factors that lead to obesity, namely socioeconomical status, lifestyle habits, sleeping patterns and genetic influence. On the other hand, they may also evaluate the possibility of an association between anthropometric parameters, such as body mass index (BMI), waist-to-height ratio or waist circumference, and the prevalence of CVD, insulin resistance, chronic kidney disease or dyslipidaemia [10,14,15]. 

Some of these studies confirm the role of an increased BMI on early markers of CVD related to childhood obesity. One example is the study performed by Twig et al., in which they analysed an Israeli cohort relating BMI at age of 6–19 years and the confirmed correlation between death and CVD later in adulthood [16,17].

Overall, previous research is focused on adulthood and long-term analysis of paediatric obesity impact. Scientific data on CV changes throughout childhood and adolescence are scarce. The study main aim is to address cardiovascular changes in children due to obesity using electrocardiogram (ECG) analysis exploration supported by data science techniques.

## 2. Materials and Methods 

With the goal to develop this research and ECG analysis, three of the most common ML methodologies were considered. Namely, Data Mining Methodology for Engineering (DMME) Applications, Sample, Explore, Modify, Model, and Assess (SEMMA) and CRoss Industry Standard Process for Data Mining (CRISP-DM [18,19].

CRISP-DM was developed by IBM in cooperation with other companies and is available as an open standard. This methodology assumes that data for the process have already been collected, which is the case for the current study [20,21].

DMME is an extension of the CRISP-DM methodology, which includes the process of data acquisition. SEMMA was developed by the SAS Institute and is more commonly used by SAS tools users, as it is integrated into SAS tools such as Enterprise Miner [18].

We decided to use the popular CRISP-DM methodology as a base, as it has already been proven successful in data mining challenges, including in the medical area [21].

Figure 1 presents the CRISP-DM model, a hierarchical and cyclic process that breaks the data mining process into the following phases. Due to page limits restrictions, the methodological phases are only briefly described. For more details, please refer to the previous bibliographic references. 

Business Understanding: To understand the project objectives and requirements, which include the problem’s definition. 

Data Understanding: To focus on becoming familiar with data and identifying data quality issues using data query, inspection and visualisation.

Data Preparation: This phase, which is also known as pre-processing, consists of selecting and preparing data, and the final dataset is the output that will be used to train and test the model.

Modelling: A model which represents existing knowledge is built through the use of several machine learning (ML) tools. 

Evaluation: Evaluate the model’s performance and utility in a way to decide if a new CRISP-DM iteration is required; otherwise, the process can proceed to the final deployment phase.

Deployment: Deployment of the solution, which includes the definition of deployment and availability plan and the final review of the process.

## 3. Cardiovascular Changes Exploration 

Following the CRISP-DM model described above as a base, several phases of the research process will be explained in detail.

### 3.1. Business Understanding

The present study aims to evaluate the existence of the cardiovascular changes, identified using ECG analysis exploration with data science techniques, in obese children at 9–11 years of age in comparison with their normal-weight counterparts. Secondarily, the study aims to recognise these ECG changes. 

With this study, we hope to provide knowledge on early cardiovascular markers that can be identified from an early age, namely from childhood, to minimise the obesity impact later in adulthood. 

In order to achieve the proposed objectives, the research team worked in collaboration with Generation XXI, a previously established prospective population-based birth cohort from Porto, Portugal, which aim to characterise prenatal and postnatal development throughout childhood, adolescence and adulthood. This cohort was established in a well-delimited geographic area in north of Portugal, with a total of 8647 children, consecutively born between April 2005 and August 2006 in one of all five public maternity units covering six municipalities of the metropolitan area of Porto. The cohort characteristics are summarised in Table 1. Throughout the sequential evaluation, all the participants were submitted to anthropometric, laboratory and cardiovascular assessment (including office BP and ECGs). The project gathered the ECGs realised at the ten-year evaluation and resorted to data science techniques to analyse the ECG changes [22].

The strategy for the data mining process consists of developing a performant machine learning model able to classify the ECG beats in obese and in normal-weight children and adolescents. With the model building phase completed, it would be possible to analyse the model’s “hidden” structure and clarify the model classification. This analysis is performed with explainable Artificial Intelligence (XAI) techniques.

Taking into consideration that classifying heartbeats from normal-weight or obese children is a machine learning classification problem, several metrics can be used for measuring the model’s performance. Some of these include accuracy, confusion matrix or several association measures, namely sensitivity, specificity, precision, recall or f-measure, receiver operating characteristic curve (ROC), and the area under the roc curve (AUC), among others. The choice of which measure to use depends highly on the problem, data, and on the objectives that the user aims to achieve. Considering all the research variables and aims, we decided to use accuracy as the evaluation/assessment metric.

### 3.2. Data Understanding

The data source of this study was directly extracted using ECG equipment that was assigned to the Generation XXI project, which was not used exclusively for this project.

The extraction of ECGs resulted in 8224 valid records in an XML proprietary vendor format. 

The information contained in the XML file was changed into a more “friendly” format, making it more suited to the data mining process. This procedure was accomplished by using the python programming language supported by the “Beautiful Soup” open source library.

The following information was extracted from the XML format:ECG collection date;Individual (and related data), including identification (encoded); birth date; weight; height; gender;ECG signal (Lead I);ECG report.

Considering the possibility of extracting data from individuals not from the Generation XXI cohort, the ECG records were analysed considering the individuals’ age on the ECG collection date. Generation XXI ECG collection was performed during the cohort follow-up visit at 10 years of age, but some children were evaluated a few months before or after the age of 10 (with a few already having completed it at 11 years of age). Thus, all the ECG collections between the ages of 9 and 11 were considered for further analysis.

Figure 2 presents the extracted age records filtered by age intervals from 0 to 15 years of age. We can observe that some ECG records were from projects other than Generation XXI.

To assure the correct processing during the next data mining phase, a complete anonymised ID list from all the Generation XXI participants was used and merged with the existing ECG extracted records in order to guarantee the inclusion of only the Generation XXI cohort participants. The final collected dataset included 4306 records from children (51% males) at 10 years of age, 917 from children at 11 years of age and 423 from children at 9 years of age. After the described steps, data collection was terminated. 

Signal extraction from the ECG records presented a sampling rate of 500 samples/s with 10 s duration, which results in a total of 5000 measures for each record.

All the ECG records were “scanned” to find duplicates, records with missing and/or abnormal signal data, which can be caused by any machinery misconnection or external noise at the ECG collection, an incorrect birth date or ECG collection time, or invalid weight and/or height data.

The signal extracted from the ECGs presented some noise, which is common in this kind of data acquisition and even more usual when it is performed on young children due to motion and respiration artefacts. It was necessary to perform ECG filtering before submitting the data to ML models processing. Figure 3 presents an example of an extracted ECG signal. 

Although feature engineering is usually a task that is performed in the data preparation phase, we decided to create the derivative feature BMI (a derivative feature is a new feature which results from original features), as it represents useful information to analyse during the current phase, including anomalies and invalid data. The BMI was calculated according to the standard formula and BMI-for-age values were classified according to the WHO reference data for the BMI z-score into the following categories: normal weight (≤+1 SD, including only one thin child), overweight (>1 SD and ≤+2 SD) and obesity (>2 SD) [23].

Following the WHO, the BMI categories considered in this study were:Female: underweight/regular weight (<19), overweight (from 19 to <22.6) and obese (greater than 22.6).Male: underweight/regular weight (<18.5), overweight (from 18.5 to <21.4) and obese (greater than 21.4).

The histogram of BMI distribution and the bar chart for the BMI categories, in males and females, are shown in Figure 4.

Figure 4A highlights the BMI distribution with gender segmentation. We can observe that both curves, despite the gender, have a Gaussian or normal distribution, with data slightly skewed to the right. The distribution shapes are quite similar. 

Figure 4B presents the frequencies by BMI category and gender. In our sample, around 22/26% were categorised as overweight, and 27/22% were categorised as obese among the males/females, respectively.

### 3.3. Data Preparation

The data preparation phase includes several tasks performed to reach the final dataset to be processed, whose quality is crucial for the success of the models’ outcomes.

The aim of this study was to determine if obesity was associated with ECG changes in the studied population. In order to reduce the noise among the data and to improve the study outcomes, we decided to compare the obese group with the normal-weight group. The records from the overweight individuals were excluded from the analysis.

Each extracted ECG record was enriched with the medical report of the exam. The report was always performed by a paediatric cardiologist in text format, summarising any alterations observed on the ECG. To focus on the study purpose, we decided to eliminate any possible source of noise caused by “external” factors by excluding from the final analysis any record reported as abnormal. The most commonly reported abnormalities were arrhythmias, such as early repolarisation patterns and bradycardia, auriculoventricular heart blocks, ventricular hypertrophy and electrical axis deviation. Following this, the team proceeded to signal preparation.

In the previous data understanding phase, we identified the need to apply ECG filtering to the signal records. The signal was processed with a finite impulse response (FIR) with a band-pass filter. 

A finite impulse response (FIR) filter in signal processing is one that settles at zero in a finite amount of time, making its impulse response (or reaction to any input of a finite length) of a finite duration. In contrast, infinite impulse response (IIR) filters have the potential to endlessly respond and may provide internal feedback, which often fades [24].

The resulting signal with a 10.000 millisecond duration was then decomposed on the beats. For achieving beat extraction, the first step was to identify R peaks in the full signal. The identification of R peaks was performed with the support of the BioSppy—Biosignal Processing in Python library, which applies the approach defined by P. Hamilton in this study “Open source ECG analysis” [25,26]. 

For the purpose of this study, we considered only “complete” beats, in which a complete beat is the range of signal composed of an R peak preceded by an interval of 100 measures, corresponding to 200 ms, and proceeded by 200 measures, which correspond to 400 ms. For a total of 600 ms, there are 300 samples where a “complete beat” corresponds to a PQRST complex. Every beat that did not fulfil these requirements, for example, because an R peak occurs near the start or end of the recording, was discarded to avoid dealing with “incomplete” heart cycles. 

The team’s analysis of the ECGs led to them obtaining the ranges of 600 ms, 200 ms before the R peak, and 400 ms after the peak. With this time frame, it is possible to ensure that the entire PQRST complex is captured and to maximise the extraction of valid dataset segments (heartbeats) for the following phases.

Figure 5 presents an example of the signal processing step. The figure shows the ECG signal after FIR filtering and the identification of R peaks in “complete” beats.

The process resulted in the extraction of a total of 14,308 complete beats, with a 600 ms duration (300 measures), which were considered valid according to the requirements defined before. Figure 6 presents an illustration of one of the extracted beats.

The last analysis was performed to enrich data understanding. The dataset beats signal was plotted on a graph (Figure 7) stratified by BMI classes (obese or normal), with the mean values and error bands being depicted.

The blue solid line represents the average signal for obese individuals, while the orange dashed line presents the average for the normal-weight children. The coloured areas around the lines represent the error band, which is limited by the standard deviation of the signal for each segment.

Via the analysis of Figure 7, some differences between obese and normal children beats seem to be evident. The charts with gender categorisation are similar, with gender not affecting the distribution. The final dataset was created, including complete valid beats, according to the BMI classes considered. The final dataset includes data from 1.101 individuals (14.308 complete beats), following the distribution presented in Figure 8. 

In summary, the inclusion criteria encompassed all the individuals from the Generation XXI project cohort of both male and female genders who had undergone diagnostic exams for cardiovascular assessment, namely an ECG at around the age of 10. To be more precise, a few participants were either 9 or 11 years old due to variations in the time frame of the data collection.

Given the specificity of this study and in order to reduce potential noise in the analysis and results, only individuals categorised through BMI as either “normal” or “obese” were included in the modelling phase. Those categorised as overweight were excluded. The research team identified the presence of cardiovascular anomalies as a potential risk to the study’s analysis and conclusions. Therefore, we decided to exclude all the individuals whose ECG reports indicated any abnormalities.

### 3.4. Modelling

This study encompasses the need for developing a model able to distinguish between the beats collected from normal and obese individuals. Considering the problem complexity in which the beat signal orientation/shape/trend is important, that is, the signal measure at each time/observation cannot be seen in isolation, we decided to develop a model supported by neural networks (NN).

Considering the challenging characteristics, in particular, the importance of observing measures not as independent elements, and the team’s knowledge, convolutional neural networks (CNN) were identified as the most promising ML architectures, namely, with the application of 1D convolutions. Still, at this point, all possible NN architectures were considered.

One-dimensional CNN was deemed the most promising approach, since it allows users to derive features from a fixed-length segment of data, which are, in this case, each of the heartbeats. The process allows users to learn from the signal data directly, without having to build explicit features.

The discovery of an optimised NN is an iterative, complex and time-consuming process, and thus, we decided to address it in detail in the following sub-chapter.

In the ML modelling process, the best practices demand users to split the final dataset into training, validation and testing data. One of the most common approaches was used, dividing the dataset into 70/20/10, respectively.

#### Modelling Architecture Optimisation

As previous mentioned, finding the best ML algorithm and, for this particular challenge, an optimised NN to fit the research data is a challenge by itself.

The NN algorithm family includes two types of hyperparameters, namely, training and structural hyperparameters. The structure includes decisions about the NN number of layers (deep), number of units in each layer, kernel size, number of filters, stride, pooling, normalisation techniques and activation functions. The training hyperparameters include decisions about the optimisers, learning rate, learning rate momentum, batch size, initial weights, epochs and early stopping patience. The existence of such a large amount of variables, not all have been mentioned, results in a nearly endless set of combinations.

For addressing the challenge of finding the best NN, namely, the best classification accuracy, we decided to perform an NN hyperparameters combination search, which is also known as Neural Architecture Search (NAS).

There are some experimental projects which have employed NAS. According to our research, the most relevant ones were the AutoKeras and DEvol Deep Neural Network Evolution projects. Still, neither used an NAS process that included 1D CNNs [27,28,29].

The research team has developed/adapted/extended new processes/methods, which, besides NN “common” architectures, also included 1D CNN NAS. Rala Cordeiro and Raimundo addressed this topic in detail, with a focus on CNN, convolutional neural networks, in a previous paper published in the MDPI *Sensors* Journal under the title *Neural Architecture Search for 1D CNNs—Different Approaches Tests and Measurements* [30].

The NAS process was performed using a machine with the following characteristics: OS: Ubuntu 18.04; Disk: 500 GB SSD; Sys Mem: 32 Gb; CPUs: AMD Ryzen 7 3600X; GPUs: NVIDIA GTX 1080 TI 11 GB + RTX 2060 SUPER 8 GB.

Several search approaches were applied during the process, namely, a search based on greedy, Bayesian, hyperband, random and genetic algorithms. The NAS setting highlights are as follows: epochs: 200; max trials: 400; generation’s number: 20; population size: 20. Search’s goal: maximise validation accuracy.

The result of the NAS process applied to this challenge resulted in the following NN (Figure 9).

The neural network structure (Figure 9), is as follows:Input layer: 300 inputs, corresponding to the length/size for every heartbeat (300 measures to which corresponds 600 ms).Convolutional block 1: Composed by 128 filters, 1-D kernel with size 3 and stride = 1, an ReLU activation function and a dropout of 0.45.Convolutional block 2: Composed by 16 filters, 1-D kernel with size 3 and stride = 1, a Sigmoid activation function and a dropout of 0.1.Convolutional block 3: Composed by 8 filters, 1-D kernel with size 3 and stride = 1, a batch normalisation layer, an Relu activation function and a dropout of 0.05.Convolutional block 4: Composed by 128 filters, 1-D kernel with size 3 and stride = 1, a Sigmoid activation function and a dropout of 0.4.Flatten layer: 38,400 neurons.Fully connected/dense block: 32 neurons.Output layer: 2 classes (normal and obese), with a Softmax function.

The training hyperparameters were as follows: optimisation function: AdamOptimizer; learning rate: 10^−3^; batch size: 200; early stopping criteria: 30 optimisation steps based on validation set; loss function: categorical cross-entropy; Weight’s initialisation: Xavier initialiser; Bias initialisation: zero.

The achieved NN presented the following performance.

Table 2 presents the confusion matrix for the validation dataset. This matrix allows a quick and complete view of the model outcomes produced for both classes, normal and obese. We can observe that among the predictions, a total of 2218 records were classified correctly (the sum of underline values), which correspond to the validation accuracy of 78% presented in the right table. Table 3 summarises the accuracy achieved for the three sub-datasets, namely training, validation and testing.

The achieved results were further evaluated during the CRISP-DM phase evaluation.

### 3.5. Evaluation 

The previous modelling phase was able to achieve a model with very interesting performances, with accuracies of 78/79% (highlight at 79% with the testing dataset) on a problem that includes the use of complex data, namely signal representation based on real data, which always have the challenges of “real life” scenarios, namely, errors, noise and an “imperfect” research environment. From a technical perspective, the results can be considered exciting.

Regarding the business goal, the model was able to classify the heartbeats with good accuracy, meaning that the beat signal provides relevant information used to differentiate both classes. In another way, we can affirm that the normal and obese group beats presented differences that can be “observed”/extracted using the model in order to classify them correctly. 

The achieved results allowed us to address the project objectives, revealing that some ECG differences exist and can be identified in obese individuals when compared to those of the normal individuals group.

Still, when analysing the validation confusion matrix, in Table 2, it is possible to verify that among the obese records, not all were correctly identified, with some being misclassified as normal. These results were discussed, taking into account both the technical and clinical perspectives. 

The current model considers the state of the individual at the ECG collection moment, and thus, there is no information about the BMI status duration. Therefore, it was not considered for the present analysis of the duration of obesity among the individuals classified as obese. The introduction of this information to the model would probably lead to an improvement in the model’s performance and allow us to extend the research results. This information was not available at the time the research was performed, but could be considered in the future.

Until recently, NN family algorithms were able to produce outcomes, but would not provide additional details on how the outcomes were built. These models were considered “black boxes”, and for obtaining more insights on how the results were produced, new models were required, resorting to different algorithm families (e.g., decision trees). These algorithms are easier to understand, but often entail considerable performance deterioration. 

In this study, a different approach was taken, maintaining the model and the model performance benefits, which consisted of exploring the model using explainable Artificial Intelligence (XAI) techniques. The application of XAI is presented in detail in the next section.

#### Explanation

Recently, the concept of XAI has gained interest from the machine learning scientific community as a way to obtain insights on how the outcomes of models are produced/generated. This interest grows with the complexity of the models and the difficulty in understanding them, which is one of the challenges of deep learning.

XAI can be resumed as the application of methods that allow humans to understand and interpret the results/outputs generated by AI models. A machine learning model no longer has to be a “black-box”, and the model decisions can be explained. 

A more formal definition for XAI could be “a suite of machine learning techniques that enables human users to understand, appropriately trust, and effectively manage the emerging generation of artificially intelligent partners” [31,32].

Considering the study characteristics, the goals to achieve, the model produced during the previous modelling phase (a CNN model), and the research conducted by the team, we decided to explore the model using the XAI method Gradient-Weighted Class Activation Mapping Plus Plus (Grad-Cam++). Grad-Cam is generally applied to images, generating heatmaps which allow the visualisation of areas with high importance on convolutional network architectures [33].

In a concise manner, the Grad-CAM uses a weighted combination of the positive partial derivatives of the last convolutional layer feature maps, with respect to a specific class score, as a way to generate a visual explanation for the class label under consideration.

Figure 10 presents a schema of the generic application of Grad-CAM++ on a CNN. This technique is usually used for image (2D data), and supports the generation of heatmaps. In this study, this method is applied to the generated 1D CNN, which produces a classification between obese and normal weight heartbeats and was adapted to generate a histogram, which is more suitable for 1D data. 

The technique generates the histograms presented in Figure 11, which highlight the most relevant signal moments of the classification of heartbeats for the two classes: plot A for the normal one and plot B for the obese one. As a way to reduce the noise and entropy in this analysis, the Grad-Cam technique was applied only to records that were correctly classified by the developed model.

With a detailed observation, it is possible to verify zones of the two classes which overlap. From a technical perspective, this means that the model classifies “looking” at some areas that are common to both classes. That is, the model can find relevant differences between the two classes in those areas. The overlapping areas, and consequently, the relevant differences between the two classes are reviewed and discussed in detail in the following section.

### 3.6. Deployment 

The current study incorporates the goal of developing a machine learning model able to classify individuals’ (in this case, children) heartbeats as obese or normal, though not with the purpose of making the model classification available for professionals, but instead as a basis for research/analysis purposes.

Despite not being the initial goal of the project, it would be possible to deploy the implemented model in a real scenario, which would allow the analysis of children’s heartbeats in order to identify subtle changes in ECGs that could be associated with the BMI status. The proposed model can be further developed to serve as a tool to monitor ECG changes in children and adolescents with obesity or other conditions.

The model created has been developed as part of a data science technique combination that allows the acquisition of information about the study research questions. Thus, this work will not result in an IT solution deployment, but instead, in research results, which will be presented in the succeeding sub-chapter.

#### Results and Discussion

This research had the goal of understanding if some cardiovascular changes in the ECG record could be found in obese children around the age of 10, and if these changes would be possible to identify using data science techniques. 

In the evaluation chapter complemented by the explanation sub-chapter, we were able to (1) confirm the existence of cardiovascular changes in obese children, in which a “smart” model was able to differentiate between obese and normal heartbeats with no other information besides the heartbeat signal; and (2) identify the heartbeat moments/zones that are more relevant for model classification, and consequently, those that represent relevant alterations. 

Figure 12 presented below starts by joining the most relevant moments for the classification of each of the two classes, merging the information presented in Figure 11 (explanation section). This representation allows us to perceive overlapping between the two classes. The bottom graph “captures” the classes’ overlapped areas, presenting the signal regions which are relevant to both classes’ classification. The overlap dimension was obtained from an average of both normal and obese classes CAM values. 

The bottom graph presented in Figure 12 is a visual representation of the research goal of identifying the regions where obese heartbeats present changes. 

Taking both Figure 7 and Figure 12 as a basis (analysed still on the data preparation phase), we probably achieved the best representation of the research outcomes, which are summarised in Figure 13.

Figure 13 joins together the result of the Grad-CAM++ technique (performed in the explanation section), identifying the regions of the heartbeat signal which the machine learning model considers most relevant to classification, and consequently, the ones that present the most significant differences, and the visual representation of the average and interval of confidence of the heartbeat signal for both classes.

With detailed observation, it is possible to match the signal moments with the typical heartbeat measures also known as PQRST. The figure also reveals several points where the classes signal crosses each other, which present a close relationship with the XAI application outcomes, as represented by the histogram bars.

## 4. Conclusions and Future Work

The research presented in this paper is due to a worldwide health challenge, the obesity pandemic and all its associated consequences, with an emphasis on cardiovascular comorbidities (e.g., heart failure and coronary heart disease).

According to the literature, there are several scientific studies on obesity with diverse perspectives, such as cardiology, risk factors, lifestyle habits and genetic role. However, these papers usually emphasise adulthood and the respective medium-to-long-term consequences. The evidence supporting early cardiovascular changes due to obesity through paediatric age is scant. 

This study aimed to characterise the possible cardiovascular changes in 10-year-old children due to obesity by using data science techniques on analysing ECGs from a Portuguese population-based birth cohort. This research allowed us to develop a machine learning classification to classify each heartbeat (collected from ECG) from normal-weight or obese children and describe the subtle cardiovascular changes in each heartbeat. The combination of techniques allowed us to verify the existence of cardiovascular abnormalities in obese children and identified the heartbeat areas/zones presenting changes. 

The data science process has included SoA techniques to best explore data and extract valuable insights. The authors highlight the use of NAS techniques on 1D data in a way to build the best possible machine learning model that is able to fit the study data and the applicability of XAI methods adapting/expanding existing techniques for 2D data to the study data, the signal in a 1D shape.

The research was performed with a real dataset which represents an interesting population, but more studies, if possible, applying the same proposed techniques to other populations are welcome and can contribute to knowledge enrichment. For instance, the combination of the same techniques applied to a population of another country would make it possible to analyse possible results similarities and/or differences.

The technique here present was applied to a medical diagnostic method based on a signal analysis. The technique could also be used in other signal based auxiliary exams, such as electroencephalography, electromyography or electrodermal activity ones. This may establish other forms of evolution of the current application of techniques. 

This study is a part of a larger research project that explores the potential of utilising data science in paediatric healthcare. Children constitute a dynamic population with unique characteristics, and this project endeavours to extract valuable insights and knowledge from existing datasets originally collected for different health purposes (other than data science analysis). The project also intends to contribute to bridging the substantial gap between the scientific outcomes related to child health and those concerning the entire population.

This specific study has yielded significant results and contributions in several domains:Childhood obesity and cardiovascular health: Cardiologists, including paediatric cardiologists, have long recognised the association between obesity and cardiovascular issues in adulthood. However, there is a notable absence of evidence regarding its impact on early ages. This study sought to answer the clinical question: does obesity cause cardiovascular changes in children? Regrettably, the findings confirm that obesity does indeed lead to cardiovascular changes in children as young as 10 years old. These findings underscore the urgency of addressing obesity from childhood, rather than treating it solely as an adult issue. This new information also raises new scientific questions, which will be addressed below as part of future work.Health signal-based data science: While most health data science research focuses on image processing, this study ventured into a different data format: physiological signals. Signal data readily available in the healthcare domain extend beyond ECGs and include resources like Electrodermal Activity (EDA), Photoplethysmogram (PPG), Electroencephalography (EEG) and Electromyography (EMG) records. The technique introduced by this research can be applied to diagnostic tools and other clinical resources based on 1D data. This study emphasises the untapped potential of existing data sources and provides guidance on knowledge extraction and result generation.Innovative approaches for 1D health signal data: Traditionally, convolutional neural networks (CNNs), explainable AI (XAI), and Neural Architecture Search (NAS) techniques have been applied primarily for 2D image data. This study took an innovative approach by adapting and developing these techniques to suit 1D signal data. This innovation opens up new avenues for researchers to explore similar directions in the realm of 1D data analysis.Enhancing/expanding CRISP-DM methodology: This study expanded the popular CRISP-DM methodology to incorporate concepts of modelling architecture optimisation and explainable AI, which were not present when the methodology was originally defined. This expansion aligns with the overarching goals of the larger project mentioned earlier. In our work, this expansion has demonstrated promising results and holds potential for application in future health data science research.

Despite these achievements, it is important to note some limitations:1D NAS and 1D XAI techniques: It is essential to acknowledge that the use of 1D NAS and 1D XAI techniques for signal data was initially perceived as a limitation since, at the time of this research, suitable techniques were either absent or scarce to address the specific requirements posed by the study.For 1D NAS, it became necessary to embark on a parallel research effort to develop and evaluate multiple potential approaches. The results of this parallel research are detailed in a separate publication titled “*Neural Architecture Search for 1D CNNs—Different Approaches Tests and Measurements*” [30].Similarly, the application of 1D XAI presented its own challenges, with several techniques required thorough analysis and testing. Ultimately, the Grad-CAM approach emerged as a viable solution to fulfil the study’s objectives.The research team is proud to have successfully transformed what initially appeared to be limitations into innovative approaches.Limited ECG leads: Due to time and budget constraints, the analysis was restricted to a single ECG lead (lead I). Future research could explore the incorporation of additional leads, potentially addressing all 12 ECG leads individually or collectively.Multimodal Data Exploration: From a technical perspective, there was a desire to explore combining the processing of ECG signal data (1D) with the same data in an image format (2D). This could provide insights into whether observing data in different formats (signal and image) can yield distinct or enhanced information, benefiting from the characteristic of deep learning for image analysis.Clinical Perspectives and Future Research: New questions have emerged based on the knowledge acquired through the initial research. Can the cardiac consequences of obesity be reversed with treatment? What are the consequences at different ages, like 5, 15 and 20 years old? It would be interesting to conduct an update of the study addressing children at different ages or, if it is feasible, to repeat the study with the same population at different ages.The study analyse was supported by the BMI classification at the time of the ECG collection. This metric is an anthropometric evaluation which is acceptable and commonly used in SoA studies, but not ideal [34].

From our perspective, these limitations should be recognised as constraints within the current study. However, they also represent areas ripe for future growth and beckon researchers to seize the opportunity to address them in future research. This is precisely why we have taken the initiative to outline potential future directions for each of these points.

In summary, this study represents a significant step forward in leveraging data science for children’s healthcare, providing valuable insights into childhood obesity’s impact on cardiovascular health and introducing innovative approaches for analysing 1D signal data. While it has made interesting contributions, there are exciting avenues for future research to explore and expand upon using the findings presented here.

## Figures and Tables

**Figure 1 children-10-01655-f001:**
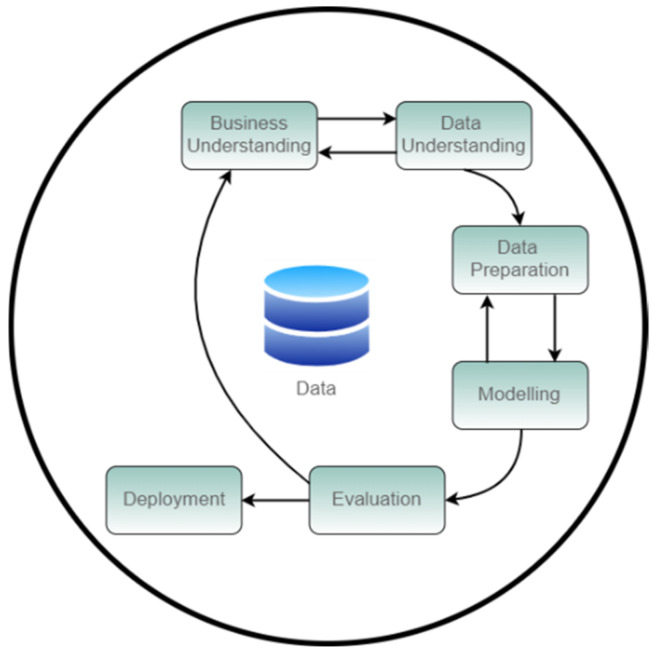
Phases of the Industry Standard Process for Data Mining (CRISP-DM) reference model (adapted from the original diagram).

**Figure 2 children-10-01655-f002:**
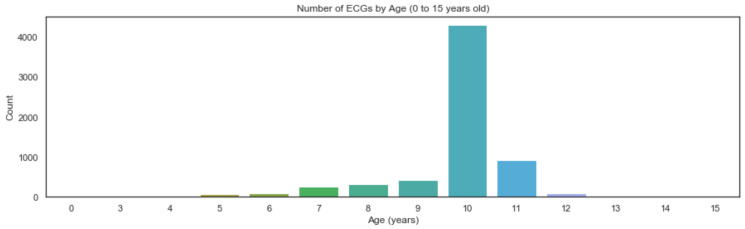
Distribution of initial ECG record extraction between 0 and 15 years of age.

**Figure 3 children-10-01655-f003:**
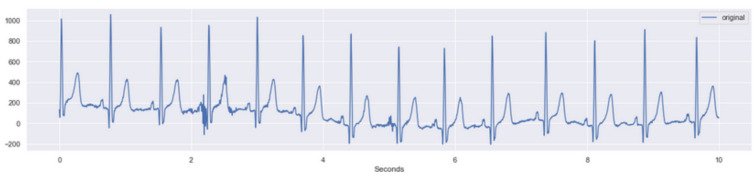
Example of a signal (the ECG lead) extract from a collected ECG.

**Figure 4 children-10-01655-f004:**
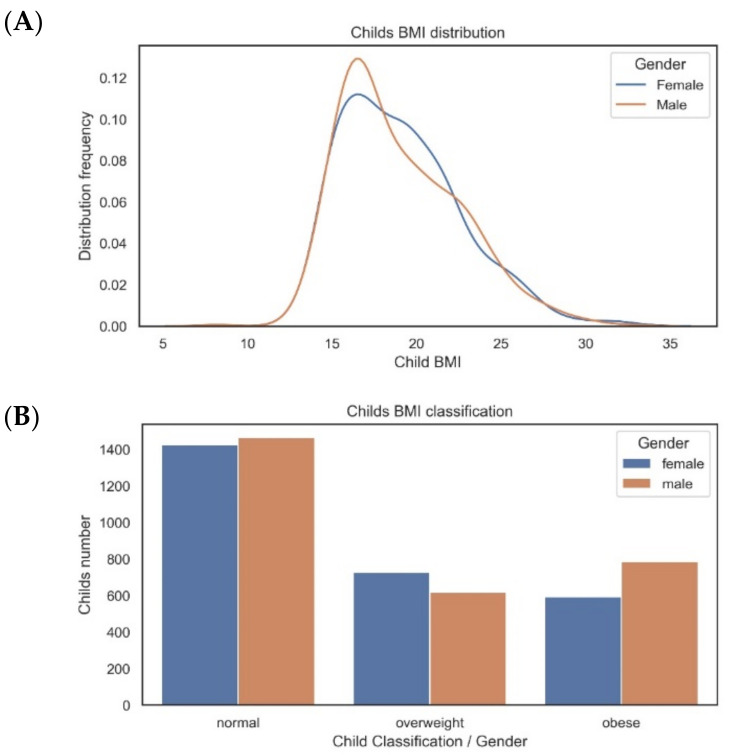
BMI statistical analysis ((**A**) distribution and (**B**) counting).

**Figure 5 children-10-01655-f005:**
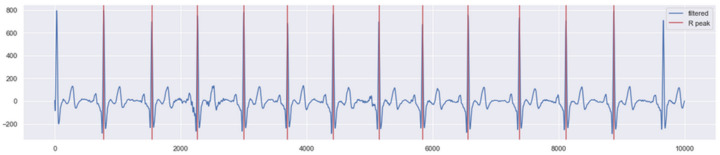
Example of ECG signal filtered with R peak and “complete” beats identification.

**Figure 6 children-10-01655-f006:**
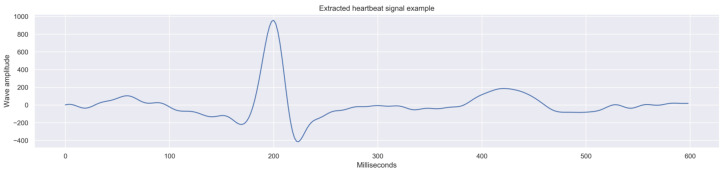
Example of a beat extract (complete) from a filtered ECG.

**Figure 7 children-10-01655-f007:**
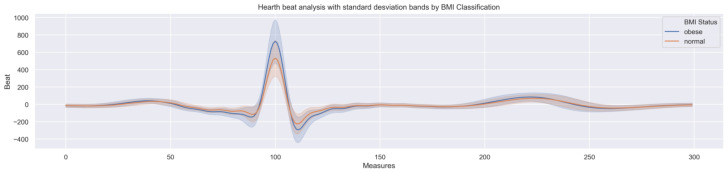
Beats signal mean and error bands stratified by BMI classes.

**Figure 8 children-10-01655-f008:**
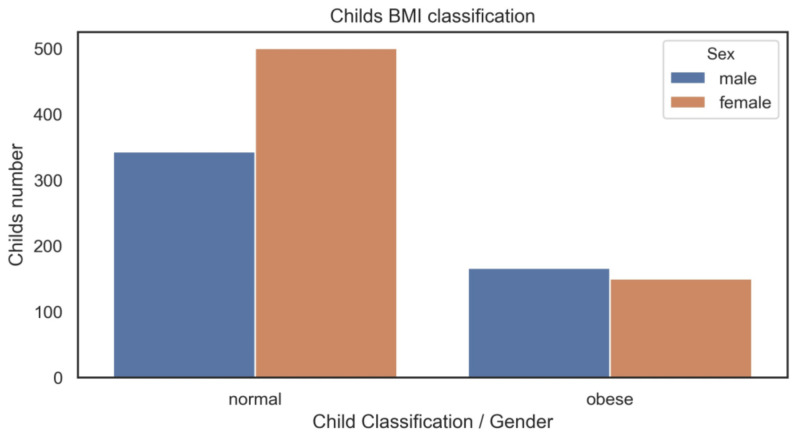
Final dataset BMI statistical analysis by gender.

**Figure 9 children-10-01655-f009:**
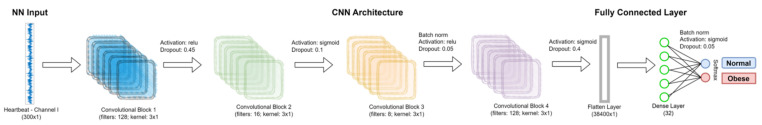
NAS process outcome, which is the best NN architecture for the research problem.

**Figure 10 children-10-01655-f010:**
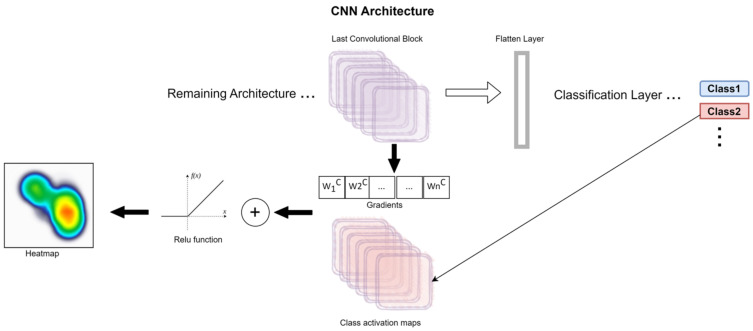
Grad-CAM++ method illustration.

**Figure 11 children-10-01655-f011:**
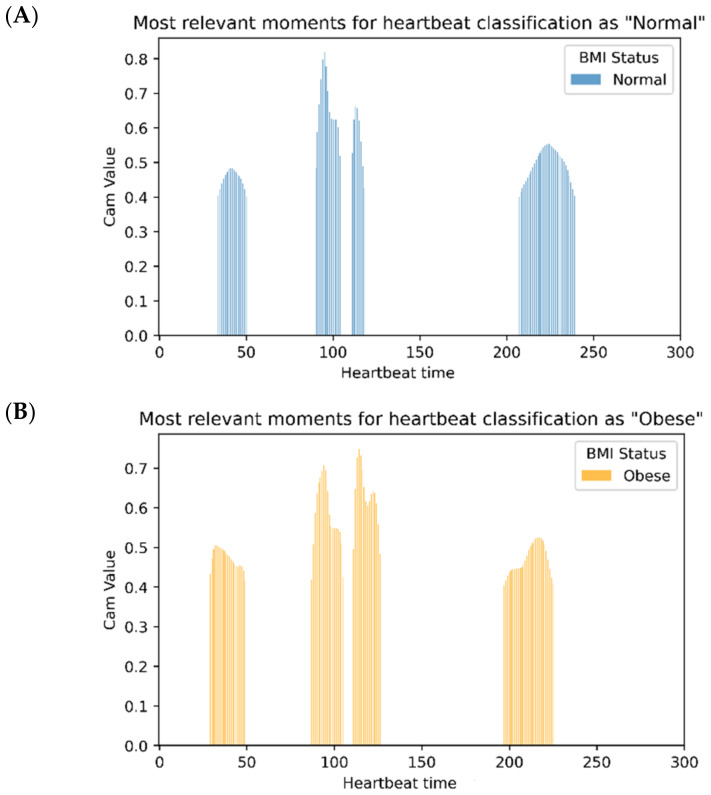
Most relevant moments for heartbeat classification using Grad-CAM++: (**A**) normal; (**B**) obese.

**Figure 12 children-10-01655-f012:**
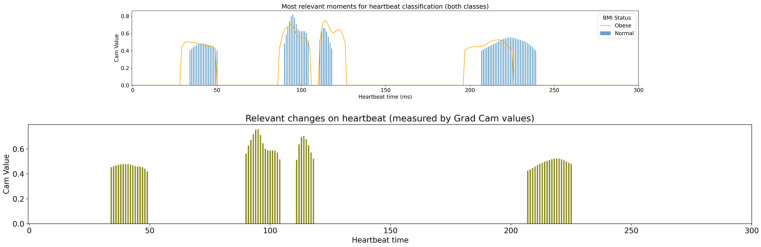
Grad-CAM analysis of both classes.

**Figure 13 children-10-01655-f013:**
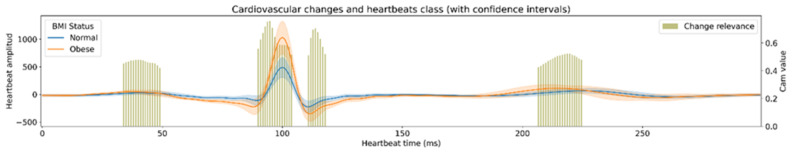
Grad-CAM results with the representation of the two heartbeat classes shapes.

**Table 1 children-10-01655-t001:** Generation XXI cohort.

Characteristics	Description
Cohort recruitment date	Abril 2005–August 2006
Localisation	Porto metropolitan area
Hospital units	Centro Hospitalar de Vila Nova de Gaia/Espinho, Centro Hospitalar do Porto—Maternidade de Júlio Dinis, Hospital de São João, Centro Hospitalar do Porto—Hospital de Santo António e Unidade Local de Saúde de Matosinhos—Hospital Pedro Hispano
Population	8647 children, born alive between April 2005 and August 2006 with at least 24 weeks of gestation 4410 (51%) males and 4237 (49%) females
Data collection process	Face-to-face interviews and diagnostic tests conducted in the hospital units and at the Generation XXI project facilities
Data collection periods	Birth, 6, 15 and 24 months, and at 4, 7, 10 and 13 years old (still ongoing)
Data collected	Various health-related parameters (social, behavioural, organisational, biological), in order to monitor population evolution. E.g.: anthropometry and blood pressure measurements, blood sampling and bioimpedance, electrocardiogram, spirometry, pubertal status evaluation and cognitive evaluation

**Table 2 children-10-01655-t002:** Confusion matrix performed on validation.

		Prediction	
		Normal	Obese
True Value	Normal	2018	99
	Obese	544	200

**Table 3 children-10-01655-t003:** Datasets classification accuracy.

	Training	Validation	Testing
Model Accuracy	78%	78%	79%

## Data Availability

Not applicable.

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
