# Peer review of "The Association between Childhood Obesity and Cardiovascular Changes in 10 Years Using Special Data Science Analysis"

_children, 2023, doi:10.3390/children10101655_

Round 1
Reviewer 1 Report
The present investigation utilizes a multifaceted approach to examine the impact of pediatric obesity on cardiovascular complications
There are some concerns needs to be addressed
1. Considering the nature of study, it is important to include the following information
a. The gender distribution of participants analyzed in the study.
b. The total number of participants enrolled in the study.
2. While the study primarily centers on the relationship between obesity and cardiovascular changes, it is essential to elucidate the factors used to define obesity among the participants. This can be achieved by including participants' BMI (Body Mass Index), which would provide clarity on this aspect.
3. To enhance the study's credibility, it is advisable to include baseline characteristics of the participants, providing a more authentic context for the research.
4. Regarding ECG measurements, it is important to elucidate the methodology for calculating beats per minute (BPM) used in the machine learning approach.
5. The inclusion and exclusion criteria employed for participant selection should be clearly outlined in the study."
Author Response
Dear Editor and Reviewer, firstly, we would like to thank the reviewers for all the constructive comments on our manuscript. In this revised version we have made additional changes in order to comply with the reviewers’ comments and recommendations, as well as other improvements that we consider relevant. The objective of this document is to briefly explain how we took the comments into account and modified the manuscript accordingly. The comments of each of the reviewers are followed by a text containing the answer to the respective comments (in italics and in blue, as in the current paragraph).
All updates carried out to improve the readability of the manuscript are highlighted through the track changes feature in the Word file. The identification of the text lines used in this current document refers to the document with change’s markups.

Reviewer 2 Report
Dear Authors,
I have carefully reviewed the manuscript , and I find that it presents valuable contributions to the field. Overall, the authors have done a commendable job in their research. Here are my comments and suggestions:
1.While the conclusion provides a broad overview of the study's purpose and methods, it lacks a specific mention of the results and their implications. Adding a brief summary of key findings would enhance the conclusion's completeness.
2.Consider including a brief section in the conclusion that discusses the limitations of the study. Mentioning any potential challenges or constraints faced during the research would provide transparency to readers.
3. While the conclusion hints at the potential for further studies in different populations using similar techniques, it would be beneficial to provide more specific suggestions for future research directions. This could help guide researchers interested in building upon your work.
4.Discuss the practical implications of your findings. How might the identification of cardiovascular abnormalities in obese children impact clinical practice or public health interventions? Addressing this aspect would add depth to the conclusion.
Author Response
Dear Editor and Reviewer, firstly, we would like to thank the reviewers for all the constructive comments on our manuscript. In this revised version we have made additional changes in order to comply with the reviewers’ comments and recommendations, as well as other improvements that we consider relevant. The objective of this document is to briefly explain how we took the comments into account and modified the manuscript accordingly. The comments of each of the reviewers are followed by a text containing the answer to the respective comments (in italics and in blue, as in the current paragraph).
All updates carried out to improve the readability of the manuscript are highlighted through the track changes feature in the Word file. The identification of the text lines used in this current document refers to the document with change’s markups.
Thank you very much.
Sincerely,
Octavian Postolache

Round 2
Reviewer 1 Report
Authors have addressed all the review points and can be accepted in this form for publication